# Critical properties of a comb lattice

Natalia Chepiga[1⋆] and Steven R. White[2]

**1** Institute for Theoretical Physics, University of Amsterdam,
Science Park 904, 1098 XH Amsterdam, The Netherlands
**2** Department of Physics and Astronomy, University of California, Irvine, CA 92697, USA

⋆ natalia.chepiga@alumni.epfl.ch

## Abstract

In this paper we study the critical properties of the Heisenberg spin-1/2 model on a comb lattice — a 1D backbone decorated with finite 1D chains – the teeth. We address the problem numerically by a comb tensor network that duplicates the geometry of a lattice. We observe a fundamental difference between the states on a comb with even and odd number of sites per tooth, which resembles an even-odd effect in spin-1/2 ladders. The comb with odd teeth is always critical, not only along the teeth, but also along the backbone, which leads to a competition between two critical regimes in orthogonal directions. In addition, we show that in a weak-backbone limit the excitation energy scales as $1/(NL)$, and not as $1/N$ or $1/L$ typical for 1D systems. For even teeth in the weak backbone limit the system corresponds to a collection of decoupled critical chains of length $L$, while in the strong backbone limit, one spin from each tooth forms the backbone, so the effective length of a critical tooth is one site shorter, $L-1$. Surprisingly, these two regimes are connected via a state where a critical chain spans over two nearest neighbor teeth, with an effective length $2L$.

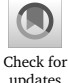 Check for updates

# 1   Introduction

Antiferromagnetic Heisenberg spin chains have been studied intensively over the years. In one-dimensional systems frustration is usually induced through competing interactions and is known to lead to various exotic phases and quantum phase transitions. The effects from competing interactions, which in 1D are often equivalent to geometric frustration, have been studied intensively within the framework of $J_1 - J_2$ chains [1–6] and spin ladders [7–11]. Alternatively, geometric frustration can be added to a system through a decoration with dangling spins [12–14], sometimes known as a Kondo necklace problem [15, 16]. Traditionally the number of decorating spins in the necklace models is limited to just a few. In the present manuscript we will focus on a so-called comb lattice where the number of pending spins is comparable to the length of the main chain - the backbone.

The simplest comb lattice is a tree lattice that consists of spin chains coupled to each other through one edge, as schematically sketched in Fig.1. A recently proposed comb tensor network [17] provides an efficient way to simulate this model numerically. The method has been bench-marked on a spin-1 Heisenberg comb lattice and has revealed a number of unusual states caused by the lattice geometry. In particular, the states include an emergent critical spin chain formed out of spin-1/2 edge states confined at the edges of the Haldane chains, and higher-order edge-states emergent at the two corners of a comb lattice [17].

Identification of the universality classes of the quantum critical lines is of central interest in one-dimensional many-body physics. In most of the cases the critical properties can be described by the underlying conformal field theory (CFT). In some rare cases, however, this is not possible due to perturbations that drive the system out of the conformal regime. Perhaps the most celebrated example is associated with a relevant chiral perturbation that leads to a continuous quantum phase transition is a non-conformal Huse-Fisher universality class [18–20]. In the present manuscript we will provide another example, in which the conformal criticality is destroyed by purely geometric frustration. This geometric frustration does not involve triangular arrangements of spins or next-neighbor couplings, which clearly cause frustration to a Néel spin pattern. Instead, within a valence bond picture it stems from the fact that a spin with three neighbors can only form a valence bond with one of them.

In the present paper we study critical properties of the spin-1/2 Heisenberg model on a comb lattice defined by the following Hamiltonian:

$$H = J_{bb} \sum_{i=1}^{N-1} \mathbf{S}_{i,1} \cdot \mathbf{S}_{i+1,1} + J_t \sum_{i=1}^{N} \sum_{j=1}^{L-1} \mathbf{S}_{i,j} \cdot \mathbf{S}_{i,j+1}, \tag{1}$$

where $J_{bb}, J_t > 0$ are antiferromagnetic nearest-neighbor interactions along the backbone and along the teeth, correspondingly. Without loss of generality we set the tooth coupling constant to $J_t = 1$. By contrast to the spin-1 case [17], the system is critical even in the absence of backbone interaction.

Unlike an infinite-dimensional Bethe lattice [21], the comb lattice is a one dimensional system. This conclusion can be approached from two different starting points. First, the comb can be viewed as a set of 1D chains – the teeth. When the backbone interaction is not relevant it can be considered as a special boundary conditions (or boundary field) applied to critical chains sitting on the teeth. In this respect, the comb lattice is a straightforward generalization of a Y-junction of three chains [22] and of a chain with an impurity bond [23] studied previously. For the latter it has been shown that the ground state of critical chains changes drastically while tuning the impurity coupling. On the other hand, the comb is a highly decorated chain – the backbone; and can be considered as a generalization of the necklace problem. In this paper we study what happens in the interplay of these two regimes when the system is critical, in particular, when it is critical in *in both directions*.

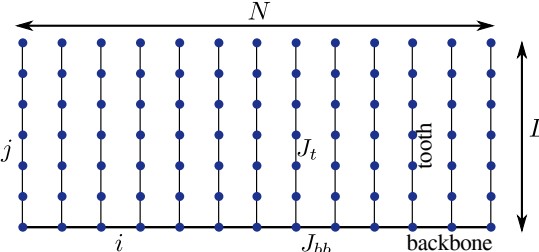

Figure 1: (Color online) Sketch of the comb lattice. Each tooth corresponds to a finite-size spin-1/2 Heisenberg chain with nearest-neighbor coupling constant $J_t = 1$. Edge spins on one edge of the tooth are coupled along the backbone with coupling constant $J_{bb}$.

In our study we will mainly focus on the three regimes: a weak backbone limit $J_{bb} \ll 1$, a competition between the teeth and the backbone $J_{bb} \approx 1$, and a strong backbone regime $J_{bb} \gg 1$. Fig.2 provides a first insight into these three cases. It shows nearest-neighbor correlations (blue) and bipartite entanglement entropy (green) on nearest-neighbor bonds. In order to compute the entanglement entropy we divide the system into two different pieces in two different ways, as shown in Fig.3, and compute the reduced density matrix $\rho$. The entanglement entropy is then given by $S = -\text{Tr}\rho \ln \rho$. In the first type of bi-partition, the system is cut across the backbone such that each tooth belongs entirely to one of the two subsystems. In this way we measure the entanglement carried by the backbone. In the second type of bi-partition, one subsystem includes a set of sites at the tip of the selected tooth, while another subsystem contains all the remaining part of the comb, as shown in the right panel of Fig.3.

In order to focus on bulk behavior, we look at a small window in the middle of the backbone of a $30 \times 30$ comb. When $J_{bb} = 0.1$ the correlations and the entanglement are concentrated within the teeth, which remain almost uncorrelated and disentangled. One sees also a dimerization that appears at the end of each tooth and associated Friedel oscillations. This is a common consequence of open boundary conditions in critical Heisenberg chains. At intermediate backbone couplings, specifically $J_{bb} \approx 1$, the nearest-neighbor correlations along the tooth and along the backbone are almost equal. Moreover, the entanglement is almost equally distributed on all the bonds not too far from the backbone. In practice, this means that cutting, say 3/4 of one tooth, or a half of the entire comb cost essentially the same amount of entanglement. In the third regime the backbone coupling is so strong that the system prefers to have as much correlation and entanglement within the backbone. As a results, the strong backbone is almost completely decoupled from the rest of the system. This naturally makes weakly coupled teeth one site shorter. In the following we will provide more details on each of these three regimes.

## 2 The weak-backbone limit

Let us first consider the limit of a weak backbone interaction. According to Fig.2 we might expect nearly decoupled Heisenberg chains on teeth. An isolated spin-1/2 Heisenberg chain is known to be described by the Wess-Zumino-Witten (WZW) SU(2)$_{k=1}$ [24] critical conformal field theory (CFT) in 1+1D. According to WZW SU(2)$_1$ CFT the singlet-triplet gap scales with the length of a chain $L$ as $E_1 - E_0 = \pi v/L$, up to some logarithmic corrections. In the limit of nearly decoupled teeth the singlet-triplet excitation on the comb lattice can also be considered as a corresponding excitation of a single tooth. In the presence of a weak inter-tooth interaction one naturally expects this excitation to be slightly delocalized, corresponding to a

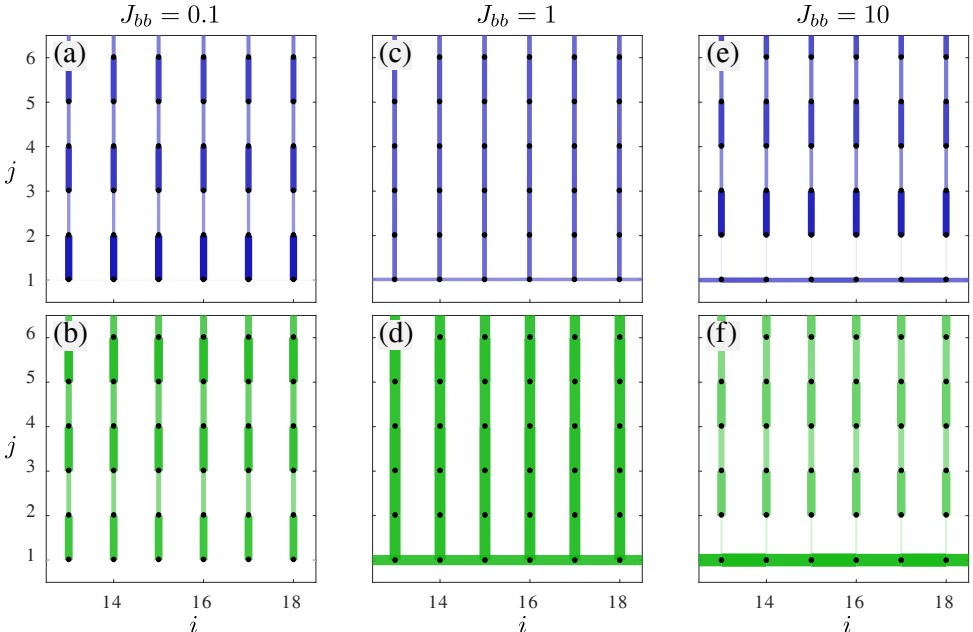

Figure 2: (Color online) Nearest-neighbor spin-spin correlations (upper panels) and the entanglement entropy (lower panels) on a comb lattice with various coupling constants on a backbone. Only a small part in the middle of the backbone of a comb with $30 \times 30$ sites is shown. The width of the lines and the intensity of the color are proportional to the strength of the correlation (upper panels, blue) or entanglement (lower panels, green).

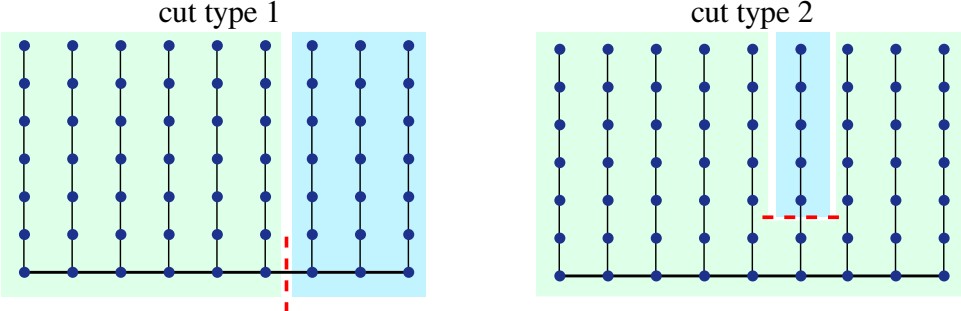

Figure 3: (Color online) Two types of bipartition used in this work to compute the entanglement entropy, background colors mark two subsystems created upon the bipartition.

finite-width soliton along the backbone. Interestingly enough, the singlet-quintuplet excitation of an isolated chain scales with its length as $E_2 - E_0 = 4\pi v/L$. It makes it energetically favorable for a comb to accommodate several spin-1 solitons before exciting a tooth to a higher state.

This picture is confirmed by our numerical results presented in Fig.4(a). By looking at the excitation energy between the lowest state in the sectors with different total magnetization we observe states with one, two and three magnetic solitons. It is spectacular that the finite-size scaling of the two- and three- solitons state are in perfect agreement with the scaling for a single soliton multiplied by a corresponding integer. In practice, this means that on a chosen scale, these solitons remain almost decoupled.

The scaling deviates from linear behavior due to the presence of logarithmic corrections

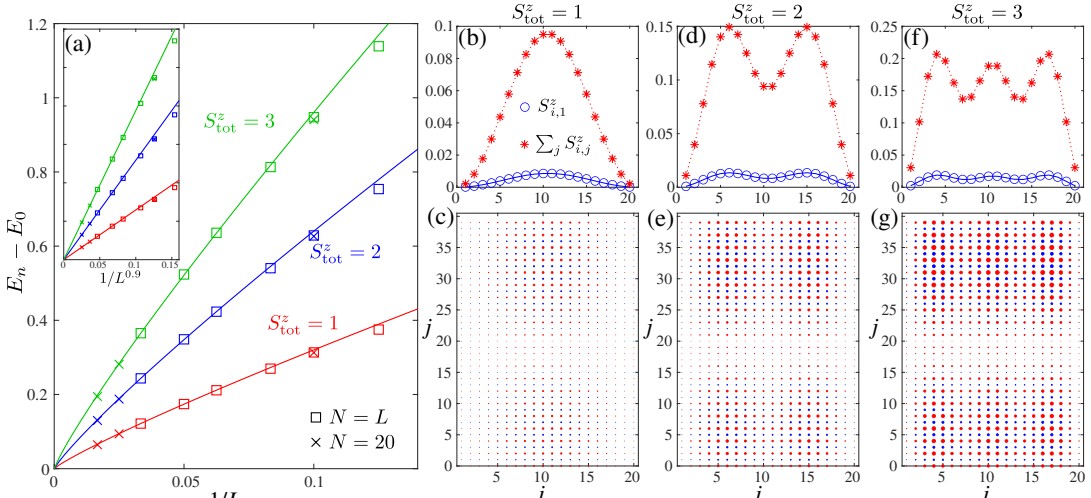

Figure 4: (Color online) Magnetic excitations in a weak-backbone regime. (a) Scaling of the energy difference between the lowest energy state in the sector of total magnetization $S^z_{\text{tot}} = 1, 2, 3$ and the ground-state in the sector $S^z_{\text{tot}} = 0$ as a function of the inverse of the tooth length $1/L$. The symbols are the DMRG data, the red line is the result of the fit of the singlet-triplet gap with $\propto L^{-d_{\text{app}}}$, and the blue (green) line is the result of the fit multiplied by an integer 2 (3). Inset: The same plot, but as a function of $1/L^{0.9}$. (b-g) The distribution of local magnetization on a comb lattice with $L = 40$ and $N = 20$. (b), (d), (f) The local magnetization on the backbone (blue circles) and total magnetization of a tooth as a function of tooth index (red stars). (c), (e), (g) Local magnetization, with the size of the circles proportional to the absolute value of the magnetization, and with red and blue colors signifying positive and negative signs of magnetization.

that for an isolated chain take the form $\propto -\frac{\pi v}{L \log L}$. Therefore, an apparent scaling dimension $d_{\text{app}}$ extracted from the fit to $E_n - E_0 = \pi v n / L^{d_{\text{app}}}$ is expected to be smaller than its true value $d = 1$. This qualitatively agrees with our finding shown in Fig.4(a) that $d_{\text{app}} \approx 0.9$. Another source of log-corrections is caused by a weakening of the boundary conditions at one edge of a tooth due to a presence of backbone interactions; although according to Fig.8(a) this boundary effect is relatively small for $J_{bb} \lesssim 0.3$.

Finally, we would like to stress the independence of the results on the number of teeth in the comb. Of course, this statement is true only when the number of solitons is sufficiently small compare to the total number of teeth; however even for three solitons allocated on a comb with $N = 8$ teeth this property holds reasonably well.

So far we have only considered a comb with an even number of sites per tooth, but the picture changes drastically when $L$ is odd. An isolated Heisenberg chain with an odd number of sites has total spin $S = 1/2$. Therefore, as soon as the backbone interaction is non-zero the spin-1/2 degrees of freedom on each tooth form a critical spin-1/2 chain. This can be detected, in particular, by looking at the Friedel oscillation profile along the backbone.

In an isolated spin-1/2 chain the open edges favor dimerization. This acts as fixed boundary conditions and induces Friedel oscillations. According to the boundary conformal field theory [25] at the critical point the dimerization scales away from the boundary as $D \propto x^d$, where $x$ is the distance to the boundary and $d$ is the corresponding scaling dimension. For a finite chain with $N$ sites the conformal transformation gives:

$$D(i) \propto \frac{1}{[(N/\pi)\sin(\pi i/N)]^d}, \tag{2}$$

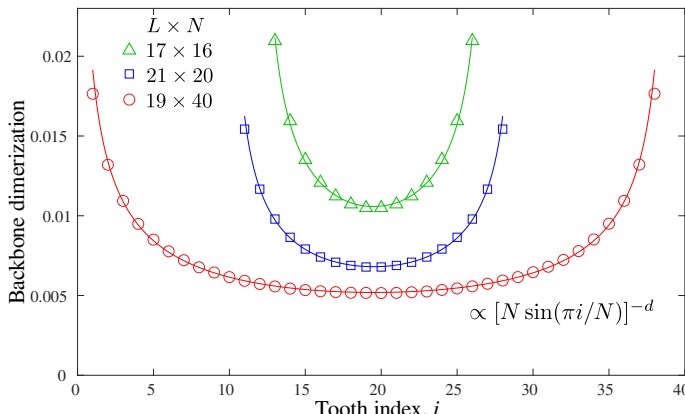

Figure 5: (Color online) Friedel oscillations along the backbone in a comb with an odd number of sites per tooth $L$. Symbols are our numerical data, lines are fit to the CFT prediction for the envelope. The resulting critical exponents are $d \approx 0.44$ for $17 \times 16$, $d \approx 0.48$ for $21 \times 20$, and $d \approx 0.51$ for $19 \times 40$, which are in good agreement with the CFT prediction $d = 1/2$.

where $D(i) = |\langle \mathbf{S}_{i-1,1} \cdot \mathbf{S}_{i,1} \rangle - \langle \mathbf{S}_{i,1} \cdot \mathbf{S}_{i+1,1} \rangle|$ is the dimerization on the backbone. In the WZW $SU(2)_k$ critical theory the dimerization is induced by a $j = 1/2$ operator with scaling dimension $d = 2j(j+1)/(k+2)$ [26]. So, for the critical spin-1/2 chain described by WZW $SU(2)_1$ an expected scaling dimension is $d = 1/2$. Fits of our data for different clusters are shown in Fig.5. The numerically extracted values of the critical exponent $d \approx 0.51$ for $19 \times 40$ and $d \approx 0.48$ for $21 \times 20$ are in a good agreement with the CFT prediction $d = 1/2$. It is important that even at the edges the dimerization is very small. This is because every spin-1/2 sitting on the backbone is mainly involved in the formation of a stronger dimer along the tooth each of which also induces Friedel oscillations perpendicular to the backbone.

Let us now look at the excitation spectrum of a comb with odd tooth length in the weak backbone regime. Quite surprisingly, the energy gap scales linearly with $1/(NL)$, as shown in Fig.6(a). This can be justified by a simple argument illustrated in Fig.7. Each tooth of a comb acts as an effective spin-1/2 degrees of freedom delocalized along the tooth. When the backbone interaction is not too big the maximum of magnetization profile on *each* tooth is approximately in its middle. So the effective distance between the nearest spin-1/2 degrees of freedom is proportional to $\propto L$, and more importantly does not change much along the chain. Then the total length of an effective spin chain is proportional to $(NL)$. According to the CFT, the energy gap scales linearly with the effective length of a critical chain, which implies $\propto (NL)^{-1}$.

According to boundary CFT an excitation energy of a chain with conformally-invariant boundary conditions scales with the length of the chain $\tilde{L}$ as $\pi n v / \tilde{L}$, where $v$ is a non-universal sound velocity and $n$ is a numerical factor associated with the energy levels that belongs to the so-called conformal towers of states. The structure of the low-energy spectra for WZW $SU(2)_k$ models with specified total spin has been worked out by Affleck et al. [26] by means of conformal field theory. Numerical calculation of *all* low-lying energy levels is often a challenging and computationally demanding task. By contrast, it has long been known that energy of magnetic excitations including singlet-triplet and singlet-quintuplet gaps can easily be obtained with DMRG by converging the lowest energy states within different sectors of U(1) symmetry, or in other words, with different total magnetization $S^z_{tot} = 0, 1, 2,...$ This method does not provide the full low-energy spectra, including multiplicities of the energy levels, but only the outer envelope. However, the special structure of the envelope is often sufficient to distinguish between various CFT candidates [6, 27]. It turns out, that for the WZW $SU(2)_1$

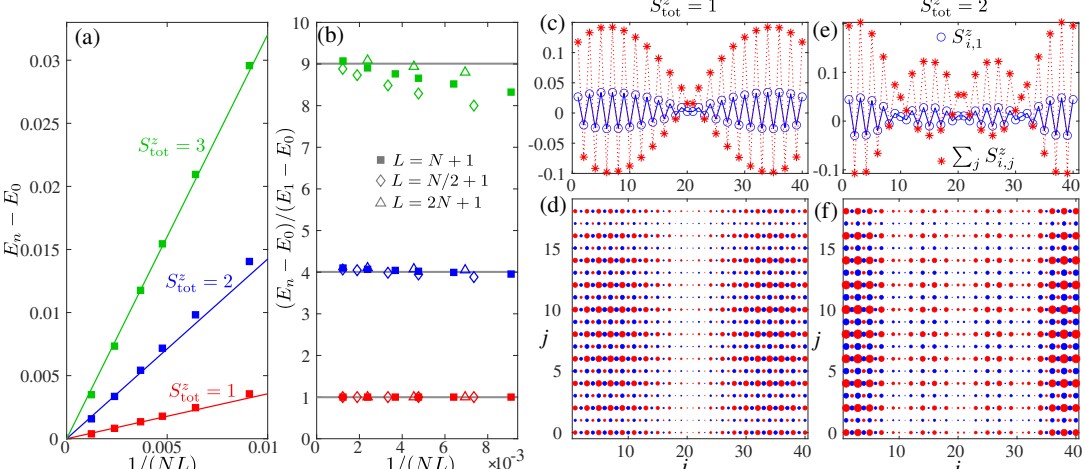

Figure 6: (Color online) Magnetic excitations on a comb lattice with an odd number of sites per tooth in the limit $J_{bb} \ll 1$. (a) Finite size scaling of the energy difference between the singlet ground-state and the lowest triplet (red), quintuplet (blue) and septuplet (green) states. Lines are linear fit of the singlet-triplet gap as a function of $1/(NL)$ multiplied by the expected structure of conformal tower $n = 1, 4, 9$. (b) The conformal tower of states extracted as a ratio of the gaps for each fixed size of the comb. Results for $n = 1$ are trivial and shown for completeness. (c)-(f) Distribution of local magnetization on a comb lattice with $L = 19$ and $N = 40$. (c), (e) Local magnetization on the backbone (blue circles) and total magnetization of a teeth as a function of tooth index $i$ (red stars). (d), (f) Local magnetization, with the size of the circles proportional to the absolute value of magnetization; red and blue colors indicate positive and negative signs of magnetization

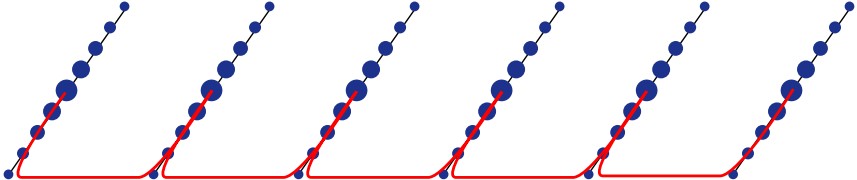

Figure 7: (Color online) Schematic representation of a state in a comb with an odd number of sites per tooth. Each tooth is in the state with $s = 1/2$, which is delocalized along the tooth. When the backbone interaction is sufficiently small, a maximal probability to find spin-1/2 is approximately in the middle of *each* tooth. So the distance between two spin-1/2 degrees of freedom is proportional to $L$ and the entire length of the effective chain is proportional to $(NL)$. Upon increasing the backbone interaction, this maximum of the spin-1/2 profile on a tooth moves towards the backbone but in a non-uniform way.

critical point with zero-spin ground-state, realized in spin-1/2 chains with even number of sites, the form of the envelope is exceptionally simple: the singlet-triplet gap scales as $(\pi v)/\tilde{L}$, the singlet-quintuplet as $(4\pi v)/\tilde{L}$, and the singlet-septuplet as $(9\pi v)/\tilde{L}$.

The results presented in Fig.6(a) for a square comb with $N = L - 1$, exhibit a good agreement with linear scaling with $1/(NL)$. This implies that in an effective CFT the linear measure of the system is given by $\tilde{L} \propto NL$. In order to check the $n = 1, 4, 9$-structure we fit the lowest singlet-triplet gap to $\pi v/(NL)$, and multiply the obtained linear function with $n = 4$ (blue line) and $n = 9$ (green) which both are in excellent agreement with the data points. In Fig.6(b)

we extract the value of the slope pre-factor $n$ numerically and compare it with the CFT $1, 4, 9$ prediction. Note that it has been obtained by dividing various magnetic gaps by the singlet-triplet gap, so results for $n = 1$ are trivial. In addition, by looking at the distribution of the local magnetization shown in Fig.6(c)-(f) one can immediately recognize the butterfly profile distinct for the critical chains and so different from the results obtained for the comb with even number of sites per tooth.

To summarize, the ground state properties of a spin-1/2 comb with even and odd teeth are fundamentally different. In a weak backbone limit $J_{bb} \ll 1$, the comb with even length teeth tends to screen the effect of the backbone interaction, while the comb with odd length teeth, as soon as the backbone coupling is non-zero, corresponds to the critical spin-1/2 chain in the direction of a backbone. Here one can intuitively rely on an analogy with spin ladders: when the number of legs is even each rung is in the $j = 0$ state; when the number of legs is odd the rungs are in $j = 1/2$ states. This gives the celebrated conclusion that spin-1/2 ladders with even number of legs are gapped, and it is gappless if the number of legs is odd. Moreover, the delocalized nature of the spin-1/2 degrees of freedom in combs with odd teeth lead to a very unusual (for one dimension) scaling of the excitation spectra - linear scaling with $1/(NL)$. The low-energy physics of combs with mixed even and odd teeth can be guessed based on these conclusions but the detailed numerical investigation of such mixed systems is beyond the scope of this paper.

# 3 Formation of double-teeth chains

Let us now look what happens to the ground state when the backbone interaction is tuned from $J_{bb} << 1$ to $J_{bb} \approx 1$ in a comb with even teeth. As shown above in the regime with small backbone coupling, the teeth are rather independent from each other and correspond to the critical Heisenberg chain described by WZW SU(2)$_1$ conformal field theory. In case of isolated chains, open boundary conditions favor dimerization and act as a fixed boundary condition, which in turn induce strong Friedel oscillations. We have already discussed the profile of the Friedel oscillation along the backbone in the context of a comb with odd teeth and weak backbone interaction. The profile of the Friedel oscillations along the isolated tooth is given by a similar expression:

$$D(j) \propto \frac{1}{[L \sin(\pi j/L)]^d}, \tag{3}$$

where $D(j) = (-1)^j \left[ \langle \mathbf{S}_{i,j-1} \cdot \mathbf{S}_{i,j} \rangle - \langle \mathbf{S}_{i,j} \cdot \mathbf{S}_{i,j+1} \rangle \right]$ is the dimerization, $1 \le i \le N$ is a tooth index, $1 \le j \le L$ is a site index within the tooth.

Fig.8(a) presents the Friedel oscillations along the middle tooth $i = N/2$ for various values of the backbone interaction for a comb with $L = 30$ and $N = 30$. For small values of $J_{bb} = 0.1$ the shape of the envelope is well captured (red symbols and line) and the scaling dimensions obtained numerically $d \approx 0.49$ is in excellent agreement with the CFT prediction. Slight increase of the backbone interaction smears down fixed boundary conditions and therefore suppress the Friedel oscillations at the edge connected to the backbone. The effect is very small for $J_{bb} = 0.3$, but starting from $J_{bb} \approx 0.5$ the deviation from the CFT profile is significant.

We noticed that further increasing the backbone interaction drives the comb though a point where the oscillation profile on a tooth resembles half of the CFT envelop on a chain with double length. In Fig.8(a) this happens at $J_{bb} \approx 0.82$. By fitting the Friedel oscillations to the (twice as big) envelop $D(j) \propto \frac{1}{[\sin(\pi(j+L)/(2L))]^d}$ we find very good agreement with our data points, although the critical exponent extracted from this fit $d \approx 0.65$ is quite far from the theoretical value $d = 1/2$.

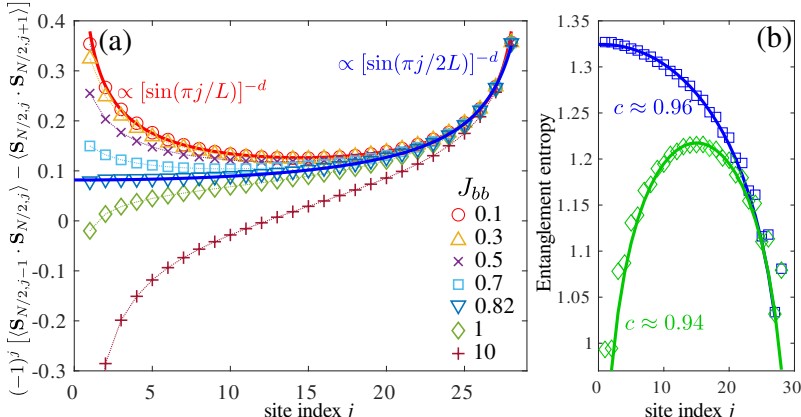

Figure 8: (Color online) (a) Friedel oscillations along the middle tooth for various backbone couplings in a comb with $L = 30$ sites per tooth and $N = 30$ teeth. Red and blue lines corresponds to the CFT fit with chain length $L$ and $2L$ correspondingly. (b) Entanglement entropy profile along the middle tooth for $J_{bb} = 0.1$ (green) and $0.82$ (blue); lines are fit to the Calabrese-Cardy formula.

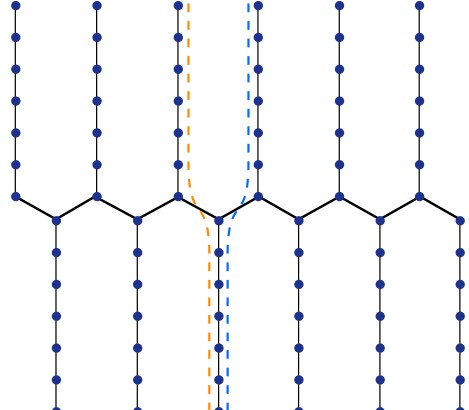

Figure 9: (Color online) Alternative representation of a comb geometry. When the backbone interaction is comparable to the interaction within the teeth, the system can be viewed as a collection of one-dimensional chains (dashed lines) with the distortion in the middle of the chain.

This observation suggests that when the coupling along the backbone and along the tooth are comparable $J_{bb} \approx J_t$ the comb resembles the collection of 1D chains extended over two neighboring teeth as sketched in Fig.9.

We can extract the central charge from the entanglement entropy profile along a middle tooth $i = N/2$. According to Calabrese-Cardy formula, in a finite-size chain with $L$ sites and open boundary conditions, the entanglement entropy scales with the size of the subsystem $l$ as [28]:

$$\tilde{S}_L(l) = S_L(l) - \zeta \langle \mathbf{S}_{i,j} \mathbf{S}_{i,j+1} \rangle = \frac{c}{6} \ln \frac{2L}{\pi} \sin \frac{\pi l}{L} + s_1 + \log g, \tag{4}$$

where $c$ is a central charge and $\zeta \approx 1$ is a non-universal constant used to suppress the Friedel oscillations [29] when the system is cut across the bond $\{(i,j), (i,j+1)\}$ and $s_1$ and $\log g$ are non-universal and universal constants. First, we benchmark the results for $J_{bb} = 0.1$. The fit of our numerical data along the central tooth to the Calabreze-Cardy formula gives a central charge $c \approx 0.94$, which is in a decent agreement with the CFT prediction $c = 1$. At $J_{bb} = 0.82$

when with the Friedel oscillations we observe the resemblance of a double-teeth chain, we find that the entanglement entropy profile also looks like a half of the profile expected for a chain with $2L$ sites. The result of the fit is in excellent qualitative and quantitative agreement with $c = 1$ profile on a chain of length $2L$.

In the simplest case of a comb with two teeth the formation of a double-teeth chain is exact and has been studied by Affleck and Eggert [23]. One can identify the following three regimes: When the coupling constant at the impurity is lower than the one in the bulk, the system renormalizes to two decoupled chains each of size $L$. When the coupling at an impurity is stronger than the coupling in the bulk, two spins connected by an impurity bond form a singlet and are effectively decoupled from the remaining chains of size $L - 1$ each. Finally, when there is no impurity, i.e. when the coupling on a selected bond is equal to the coupling in the bulk, the system is equivalent to a chain with $2L$ sites. In a comb with multiple teeth, restoration of the $2L$ chain is not expected to be exact, since each spin located at the backbone has a coordination number three and not two. That is why the agreement with the CFT profiles for chains with $2L$ sites shown in Fig.8 is impressive. As a final remark, let us point out that upon further increase of the backbone interaction we observe the third regime discussed by Affleck and Eggert: the backbone chain is decoupled from the teeth each of which is one site shorter than before. In particular, it implies that the Friedel oscillation profile for $J_{bb} = 10$ shown in Fig.9(a) is antisymmetric as in the chains with odd number of sites.

# 4 Excitations at $J_{bb} = 1$

Now lets us take a closer look at the nature of the excited states when the backbone and tooth coupling are comparable. For simplicity we take $J_{bb} = J_t = 1$. In order to extract the energy gap we compute the energy of the ground-state and the lowest energy state in the sector of $S_{\text{tot}}^z = 1$ ($S_{\text{tot}}^z = 3/2$ for $N, L$ odd). The results obtained for various values of $N$ and $L$ are summarized in Fig.10.

For a comb with even number of sites per tooth $L$ the energy of magnetic excitations scales to zero almost linearly with $1/L$ and shows fairly small finite-$N$ dependence. Note, that the even-odd-$N$ effect is negligible in this case. In order to understand the nature of these excitations we plot local magnetization profiles for both $N$-even (Fig.11) and $N$-odd (Fig.12). For completeness we also include the profiles in the weak- and strong-backbone limits.

For $L$-even and in the weak backbone limit we observe a butterfly profile characteristic of isolated critical spin chains with even number of sites. Notice on Fig.11(c) and Fig.12(c) that the butterfly is not symmetric. This can be understood by looking at the magnetization profile on a chain with $2L$ sites and a weak-bond impurity. The profiles for impurity coupling have been obtained with the standard DMRG [30, 31] and are summarized in Fig.13. Indeed the profile shown in Fig.13(c) looks similar to the profiles observed along the teeth in a weak-backbone regime in Fig.11(c) and Fig.12(c). We also stress that these excitations are well localized in the backbone direction as shown in Fig.11(b) and Fig.12(b), which agrees with our picture of solitons.

The profile changes significantly at $J_{bb} = 1$. Along a tooth the butterfly structure is replaced by its half, which supports the formation of a critical chains over two-consecutive teeth discussed above. The finite-$N$ scaling shown in Fig.10(b) suggests that for any fixed and finite value of $L$ the gap, and therefore the correlation length along the backbone remains finite. This implies that the profile observed in Fig.11(b) and Fig.12(b) will look like a soliton on a large-$N$ scale. In the present case, however this soliton is perturbed by boundaries.

Let us now look at the case of $L$-odd and $N$-even. In the weak-backbone regime, each tooth is in a state with total spin-1/2, so along the backbone we observe a critical spin-1/2 chain

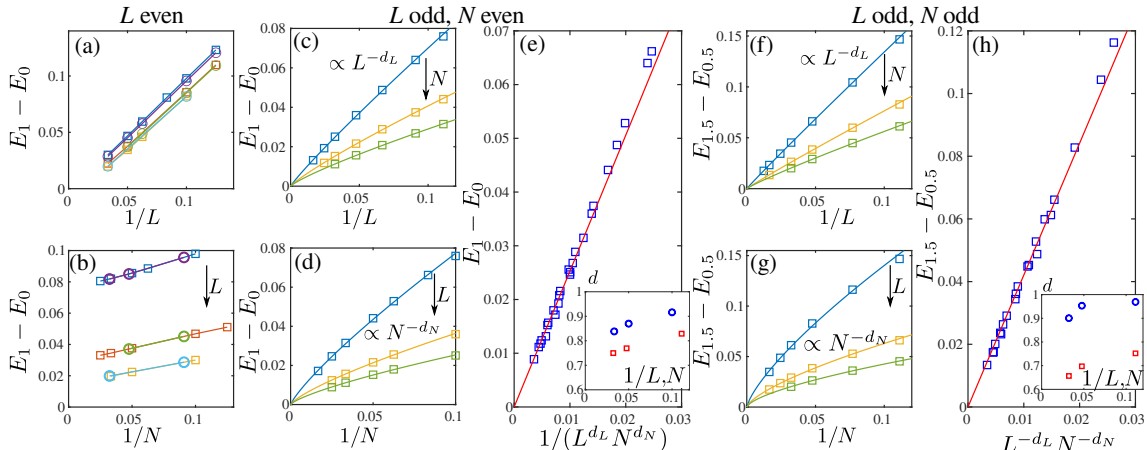

Figure 10: (Color online) Finite-size scaling of energy gaps to the lowest magnetic excitation at $J_{bb} = 1$. Finite size scaling is shown separately as a function of $1/L$ in (a),(c),(f) and as a function of $1/N$ in (b),(d),(g). In (a)-(b) we show results for $L$-even and $N$ either even (circles) or odd (diamonds). Panels (c)-(e) show results for $L$-odd and $N$-even, while panels (f)-(h) are for both $N, L$-odd. The lines in panels (c),(f) are results of a fit of the form $\propto L^{-d_L}$, and in panels (d),(g) of the form $\propto N^{-d_N}$. Panels (e) and (h) show the data collapse for the best available pair of $(d_N, d_L)$. The values of $d_{N,L}$ are summarized in the insets of panels (e) and (h). The different colors on panels (a)-(d),(f)-(g) correspond to different widths/lengths of a comb.

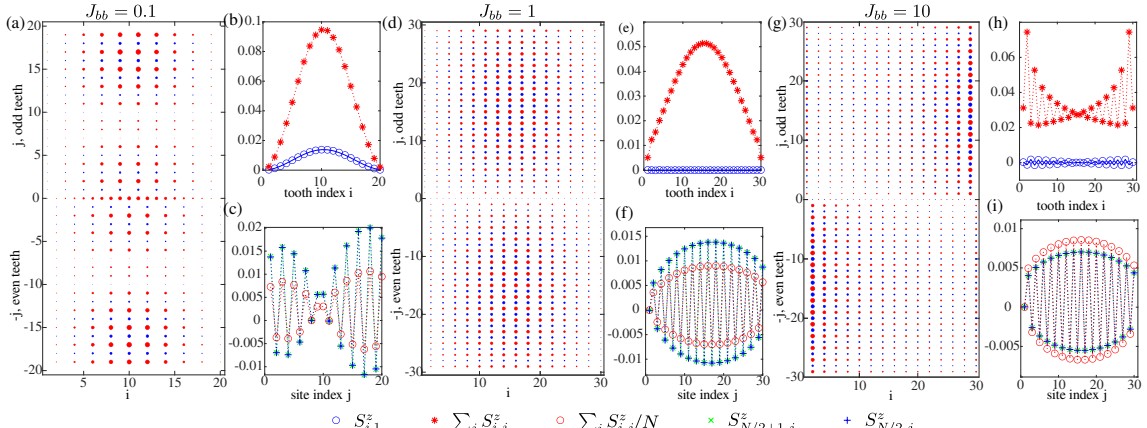

Figure 11: (Color online) Local magnetization profile on combs with even $L, N$ for three different values of backbone interaction: (a)-(c) $J_{bb} = 0.1$ and $L = N = 20$; (d)-(f) $J_{bb} = 1$ and $L = N = 30$; (g)-(i) $J_{bb} = 10$ and $L = N = 30$. In (a),(d),(g) the comb is unfolded as in Fig.9; backbone sites are placed at the coordinates $(i, j = 0)$; the radius of circles is proportional to the absolute value of local magnetization; red and blue colors indicate positive and negative sign of magnetization. Panels (b),(e),(h) show local magnetization on the backbone (blue circles) and total magnetization of teeth (red stars). Panels (c),(f),(i) show local magnetization along two consecutive teeth in the middle of a comb (green and blue crosses), and average magnetization profile along the teeth (red circles)

with a pronounced butterfly profile of local magnetization shown in Fig.14(b). Importantly, the location of the maximum (or minimum) of magnetization on different teeth is almost

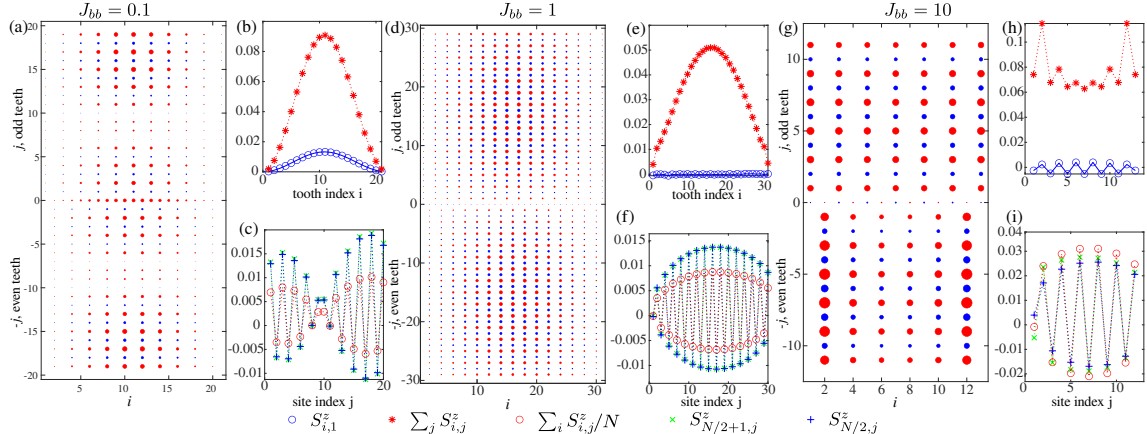

Figure 12: (Color online) Same as Fig.11 but for $L$-even and $N$-odd: (a)-(c) and $L = 20, N = 21$; (d)-(f) $L = 30, N = 31$; (g)-(i) $L = 12, N = 13$

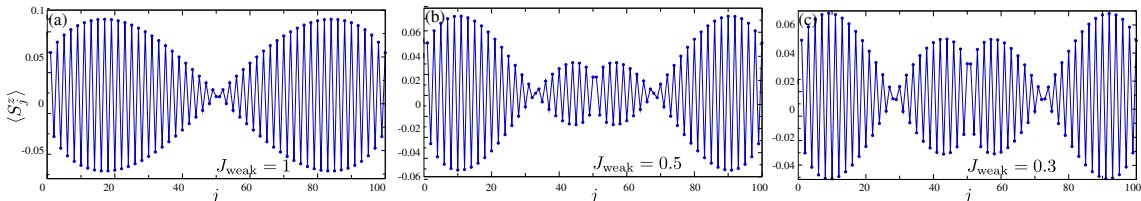

Figure 13: (Color online) Local magnetization along a finite-size chain at various values of the coupling constant at the middle bond. (a) When the coupling is uniform along the chain, the magnetization profile has a butterfly shape. When the coupling on the middle bond is smaller, the double-butterfly structure emerges. The shape of the magnetization profile at each half chain - non symmetric butterfly - is similar to the profile observed in a comb.

the same, although the values at the maximum are very different (see Fig.14(a),(c)), which qualitatively agrees with our picture sketched in Fig.7 with the uniform lattice spacing of an effective spin-1/2 chain. By contrast, at $J_{bb} = 1$ the location of maxima on different teeth are different (see Fig.14(d),(f)). So the distance between the effective spin-1/2 degrees of freedom is not uniform, as sketched in Fig.15. As a result, the finite-size scaling of the energy gap is not linear neither with $1/L$ (Fig.10(c)), nor $1/N$ (Fig.10(d)), nor with the product of the two. Moreover, the larger $N$ and $L$ we take the more freedom (or disorder) we add to the system, so less conformal the scalings are. This agrees with the fact that both values $d_N$ and $d_L$ move away from the CFT-invariant dimension $d_{N,L} = 1$. In a critical 1D system, spin-spin correlations decay with the distance between the spins as a power-law; therefore it is natural to expect that the non-uniform distribution of the spin-1/2 degrees of freedom leads to a non-uniform coupling constant in an effective spin-1/2 chain.

The same type of argument can be applied to a comb with both $N$ and $L$ odd. At $J_{bb} = 0.1$ effective spin-1/2 degrees of freedom are equally distant from each other as can be deduced from Fig.16(a). However, the equidistance is destroyed at $J_{bb} = 1$ (see Fig.16 (d),(f)) and the finite-size scaling of the gap is non-linear (Fig.10(f),(g)).

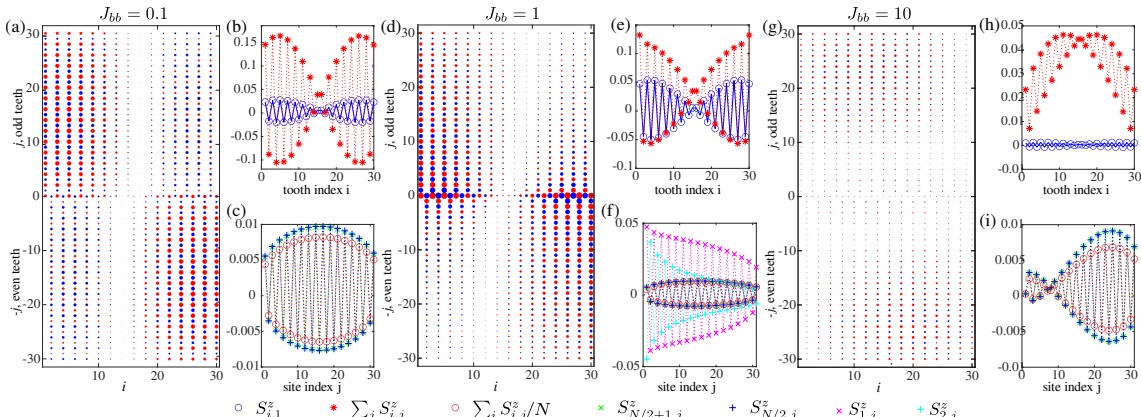

Figure 14: (Color online) Same as Fig.11 but for odd $L = 31$ and even $N = 30$. In panel (f) in addition to the average magnetization and central teeth profile, we also show the results on the first and second teeth to highlight that the profiles change along the backbone. In panel (g), with respect to panels (a) and (d), the size of circles were enlarged by a factor of 2.

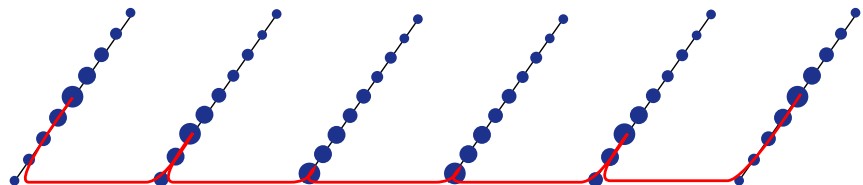

Figure 15: (Color online) Schematic representation of a state in a comb with odd number of sites per tooth and $J_{bb} \approx 1$. Spin-1/2 degrees of freedom are delocalized along the teeth in a non-uniform way, so the emergent spin-1/2 degrees of freedom are not equally spaced. This is effectively equivalent to the spin-1/2 chain with non-uniform coupling constant $J_i \neq$const.

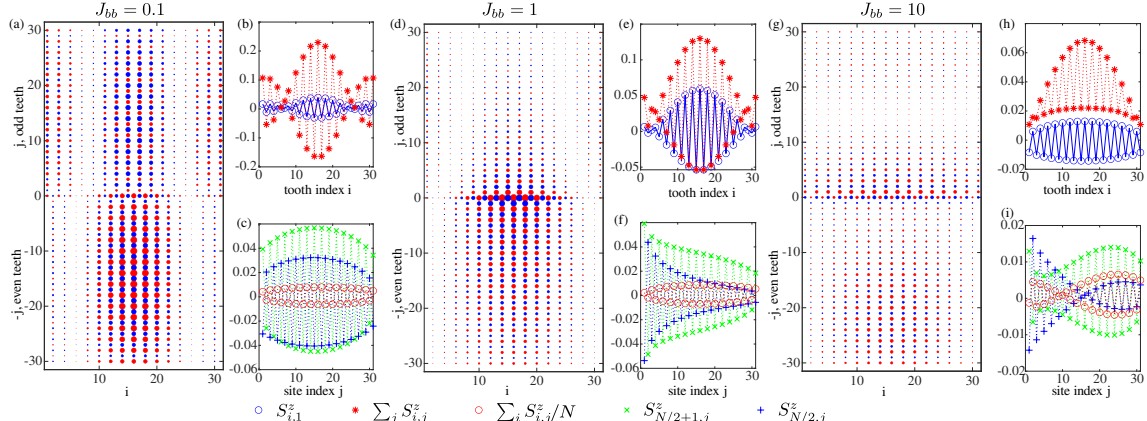

Figure 16: (Color online) Same as Fig.11 but for odd $L = N = 31$. Here we plot the difference in magnetization between the lowest energy state in the sector of $S_{\text{tot}}^z = 3/2$ and the ground-state in the sector of $S_{\text{tot}}^z = 1/2$. In panel (g), with respect to panels (a) and (d), the size of circles were enlarged by a factor of 2.

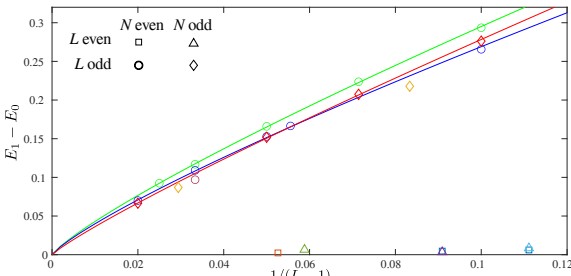

Figure 17: (Color online) Scaling of the lowest magnetic gap in a strong-backbone regime. Very low excitation energy (of the order of $10^{-3}$) is detected for $L$ even (squares and triangles). For odd $L$ the finite-size gap is much larger, but vanishes in the thermodynamic limit. Lines are the result of the fit of equal-$N$ data to $\propto (L-1)^{-d_L}$. The resulting values of the scaling dimension are within the range $0.83 < d_N < 0.89$.

## 5   Large-backbone limit

Finally, let us consider the strong backbone limit $J_{bb} \gg 1$. According to Fig.2(e)-(f), when the backbone interaction is sufficiently large, it is almost decoupled from the remaining teeth, each of which becomes one site shorter. As we know from the discussion an even-odd effect plays a crucial role. So let us start with $L$ even. The shortening of the teeth is confirmed by the anti-symmetric profile of the Friedel oscillations particularly for chains with an odd number of sites, but also observed in a comb with even $N = L = 30$ at $J_{bb} = 10$ (see Fig.8(a)). As the result, each tooth is in a state with total spin-1/2, weakly coupled to the strong backbone. The entire picture resembles the story of a two-leg ladder, which remains in the rung-dimer phase for any value of leg coupling constant. However the absence of the coupling along the "second leg" of the ladder implies that dangling spins can be polarized at very low energy. On various clusters from size $8 \times 8$ up to $20 \times 10$, for which we could reach the convergence, the gap is of the order of $10^{-3}$ (see Fig.17).

The structure of the lowest excitations can be deduced from Fig.11(g)-(i) and Fig.12(g)-(i). The triplet state corresponds to the two excited teeth far apart from each other. Interestingly enough, for both - even and odd $N$ - the maximum of magnetization is allocated at the second and the $N-1$'s teeth. For some reason these excitations avoid the first and the last teeth when $J_{bb} \gg 1$.

In case of odd $L$ the system corresponds to the critical backbone chain which is almost decoupled from the teeth with even $(L-1)$ number of particles. By analogy with the previous even-$L$ case, this can be viewed as a critical chain decorated by spin-0 objects. Because of the difference in couplings, it is energetically favorable to accommodate a triplet excitation on a tooth than to excite the "heavy" backbone. However, since the backbone itself is critical the excited tooth is delocalized as shown in Fig.14(g)-(i). The scaling of the excitation energy as in the case of an isolated chain is affected by the logarithmic corrections of the form $\propto -\frac{\pi v}{(L-1)\log(L-1)}$, which reduce apparent scaling dimension obtained from the numerical fit to $E_1 - E_0 = \pi v/(L-1)^{d_{\text{app}}}$, from its CFT value $d = 1$.

The picture is a bit more complicated when both $L$ and $N$ are odd, since the backbone itself is in the spin-1/2 state, which couples to a triplet excitation on a tooth and gives complicated structure shown in Fig.16(g)-(i).

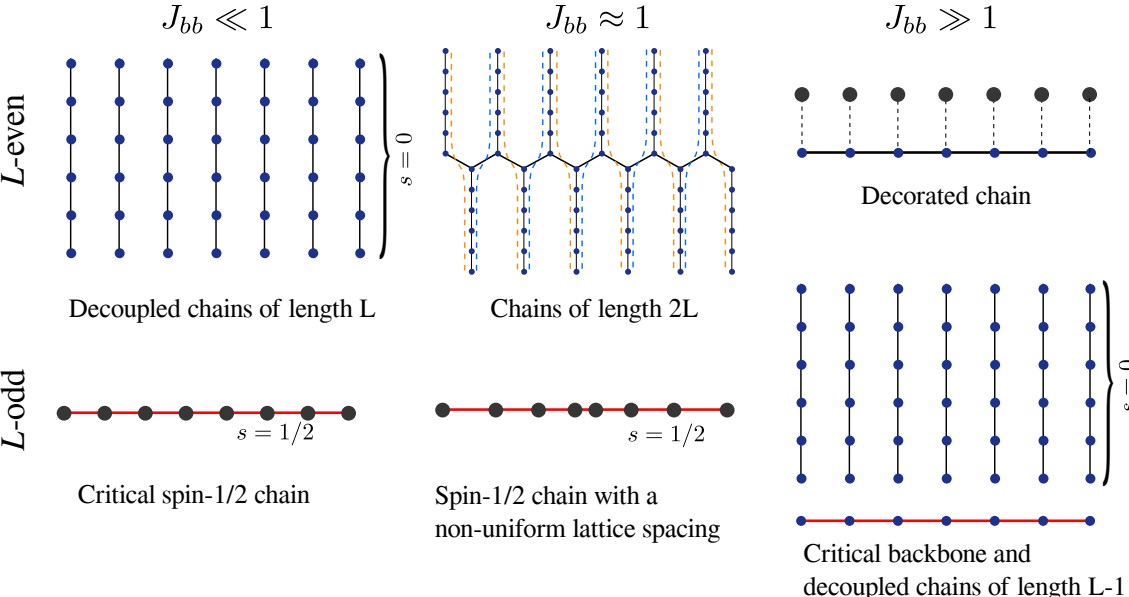

Figure 18: (Color online) Summary of the main critical regimes as a function backbone interaction and total number of spins per tooth. Non-uniform lattice spacing for $L$-odd is equivalent to the non-uniform coupling constant $J_i$ that varies along the chain.

## 6 Discussion

To summarize, we have studied numerically the Heisenberg spin-1/2 model on a comb lattice. We have found very different properties of the low-energy states depending on whether the number of sites per tooth is even or odd. The observed even-odd effect is similar to that of spin-1/2 ladders with even and odd number of legs [7].

In each case we detect three main regimes while tuning the backbone interaction. They are summarized in Fig.18. When $L$ is even, by changing the backbone interaction one can interpolate between nearly-decoupled chains of length $L$ to an extended chains of length $2L$ when backbone and teeth couplings are competing. Finally, in the strong-backbone limit the system corresponds to a two-leg ladder with zero-coupling along one leg or to the decorated spin-1/2 chain. The decorating spins are spin-1/2 degrees that corresponds to the ground-state of teeth that contains odd $L-1$ sites each.

For odd $L$ the system corresponds to the critical chain with very unusual $1/(NL)$ finite-size scaling of the spectrum due to delocalization of the spin-1/2s along the teeth. Tuning the backbone interaction effectively changes the lattice spacing in an effective spin chain. When the backbone interaction is comparable to the coupling along the teeth the effective spins are placed along the chain in a non-uniform way so the conformal invariance of the system in its usual sense is destroyed. However this opens an important question for conformal field theory: how the two CFTs in 1+1D interact with each other; and how can one describe a resulting effect of competing criticalities? In the present case the competition was induced by the chosen geometry of the lattice and naturally leads to the competition between the two CFTs. However, a similar scenario can be expected also on 2D lattices if rotation symmetry is spontaneously broken, e.g. in case of helical or stripe states close to or at the critical point. The answer to these questions lies far beyond the scope of this manuscript, however we hope that the present work will stimulate further theoretical studies in this direction.

# 7 Acknowledgements

We acknowledge insightful discussions with Ian Affleck. This work has been supported by the Swiss National Science Foundation and the U.S. National Science Foundation through DMR-1812558, and the Simons Foundation through the Many-Electron Collaboration. The calculations have been performed using the computing facilities of the University of Amsterdam.

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
