# Peer review of "Critical properties of a comb lattice"

_SciPost Physics, doi:SciPost Phys. 9, 013 (2020)_

## Round 1 · Referee Report · Anonymous · 2020-4-1

Strengths

1-Interesting model interpolating between various critical limits
2-Very thorough study containing a large number of reliable numerical data
3-Good analysis by comparing with predictions in the conformal limit

Weaknesses

1-Explanations on how certain quantities are precisely defined and calculated and how some of the symbols in the equations are defined are sometimes missing
2-Some of the CFT results appear without references.
3-Some of the panels are very hard to read because the symbols are very small and the legends are crammed into a corner

Report

The authors study a comb lattice of spin-1/2 coupled by
nearest-neighbor Heisenberg interactions. This model can be seen as a
generalization of spin systems studied previously in the literature
such as Y-junctions and spin chains with modified bonds. The results
are based on numerical data obtained using a tensor network adapted to
this geometry. This tensor network has been recently introduced by the
same authors (Ref. [17] in the manuscript). The study is very
thorough, presents a wealth of data, and analyzes them based on
theoretical predictions in the conformal limit. The results are quite
interesting: a number of critical regimes are identified. Before the
manuscript can be published, the following issues should be addressed:

Requested changes

1) Definitions of the entanglement entropy and, in particular, on how the lattice is cut exactly to obtain the entanglement entropies in Fig. 2 should be given.
2) Fig. 3 (c,e,g) and similar figures are difficult to read because the symbols are extremely small. Perhaps these figures can be improved?
3) It seems that no references are given with regard to the Friedel oscillations, e.g. around Eq. (2). Such references should be added.
4) Above Eq. (3) the authors say 'Here we expect ...'. It is not clear to which case the authors are refering here.
5) Eq. (4): The quantities in the equation should be explained, i.e., what is 'n', 'd(n)', 'g'?
6) Fig. 9: Instead of panel (i) - which does not exist - it should read panel (h) [twice].
7) There are a several typos which should be corrected.

---

## Round 1 · Referee Report · Anonymous · 2020-4-22

Strengths

1- A very detailed numerical study of a model with a geometry that allows for precise DMRG calculations.
2- Provided insight into a model that interpolates between a 1D and 2D problem.
3- The authors provide a clear understanding of the different regimes of the model, including a comparison with the conformal theory predictions.
4- An interesting crossover region was identified.

Weaknesses

1- The model itself is a kind of artificial.

Report

It is a nicely prepared manuscript about a numerical study of a spin-1/2 model with a particular geometry. While probably difficult to realize in experiments, the geometry allows for a precise numerical study of the model using DMRG. Different regimes have been identified, and the overall picture follows what we expect from the physics of 1D spin chains once the numerical results are properly interpreted. The critical correlations have been critically compared to expressions from conformal theory.
I would like to recommend the publication of the paper.

In addition to the report, a remark: it appears to me that geometry of the model allows a solution by Wigner-Jordan transformation for the XY anisotropic case - is this so, and would it be useful?

Requested changes

Fig. 17: It is not clear to me what does the "chain with a non-uniform lattice spacing" refers to. Does it mean that the couplings are different on the different bonds.

In the paragraph starting "According to Affleck et al. [25], the structure...", what is the "n=1,4,9..." ? Please explain.

---

## Round 1 · Referee Report · Anonymous · 2020-5-11

Strengths

1) High quality of data for all relevant regimes in parameter space

2) Testing of new method

2) Profound analysis of data combined with intuitive pictures

3) Very clearly written

Weaknesses

1) No discussion of possible experimental realizations

2) Deviations from conformal limit not quantitative analyzed

Report

In this paper the authors study the critical properties of the Heisenberg spin-1/2 model on a comb lattice using a newly developed comb tensor network method which they have introduced themselves in a previous publication. The model is well adapted to the strength of the method and large data sets for all regimes of interest are presented. Concurrently, it provides insights into the crossover of single impurity models such as y-junctions and non-trivial boundaries to the Kondo necklace model. The authors investigate the scaling of excited energy levels and the central charge with system size as well as the dimerization and magnetization along the backbone of the comb and the teeth. A comparison is made to CFT predictions. Particularly, there is an interesting effen-odd effect depending on the number of teeth sites and backbone sites. Besides the thorough analysis of the data this manuscript provides very convincing intuitive pictures of the different regimes and related scaling depending on the strength of the backbone coupling. My only criticism relates to the relatively short discussion of deviations from these simple pictures. In particular, the introduction of the effective exponent in the scaling as a result of the logarithmic corrections usually present in the Heisenberg chain is rather abrupt. Would it be possible to tune the disturbing marginal operator away such as in the Heisenberg chain with longer-range couplings? If the discussion of the corrections to the underlying CFT would be more substantiated I would highly recommend this manuscript for publication.

Requested changes

1) Include a more detailed discussion of the deviations of the CFT predictions, in particular give more details on the introduction and derivation of the effective exponent used in the fits.

2) In some figure legends for the magnetization profiles the indices i and j seem to be interchanged.

3) There are several typos.

---

## Round 2 · Referee Report · Anonymous (Referee 2) · 2020-6-25

Report

I would like to recommend the publication of the revised manuscript. Regarding my questions about the possibility of applying a Jordan-Wigner transformation, there is some literature about JW transformation in tree graphs, see e.g. S. Backens et al.Scientific Reports 9, 2598 (2019).

---

## Round 2 · Referee Report · Anonymous (Referee 3) · 2020-7-17

Report

The authors replied carefully to all three reports and the changes they have made in their resubmitted manuscript are convincing. I therefore recommend the publication of this manuscript in SciPost Physics as it is. Only a small remark: In Figs. 4b) and 6c) the indices i and j of the local magnetization on the backbone still seemed to be interchanged.

---

## Round 2 · Author Response

We would like to thank to all referees for their careful reading of the manuscript and for their relevant comments that allows us to improve the manuscript. Below we address questions raised by referees 2 and 3.

  1. Referee 2: Fig. 17: It is not clear to me what does the "chain with a non-uniform lattice spacing" refers to. Does it mean that the couplings are different on the different bonds. The way we observe this on a comb lattice is really a non-equally separated degrees of freedom, as sketched in Fig.15. However, an effective interaction between these degrees of freedom depends on an effective distance between them. Very intuitively, we can expect it to be proportional to the spin-spin correlations, which is in critical 1D systems know to decay with the distance between spins as a power-low. So the referee is right, the system can effectively be described by a chain with the non-uniform coupling constant J. A corresponding sentence has been added to the manuscript.

  2. Referee 2: In addition to the report, a remark: it appears to me that geometry of the model allows a solution by Wigner-Jordan transformation for the XY anisotropic case - is this so, and would it be useful? Exact solutions are always useful, of course. However, I do not see how Jordan-Wigner transformation can be applied here when both N and L are macroscopic? The coupling along the backbone causes the same problem as Jordan-Wigner transformation in 2D, the only difference is that there is only L problematic terms for a comb instead of LN as for 2D lattice. If I miss the point, further comments are welcome.

Referee 3: In particular, the introduction of the effective exponent in the scaling as a result of the logarithmic corrections usually present in the Heisenberg chain is rather abrupt. Would it be possible to tune the disturbing marginal operator away such as in the Heisenberg chain with longer-range couplings?

3.We have significantly extended the discussion of log-corrections. The notion of an apparent scaling dimension has been properly introduced, and its deviation from the CFT value caused by the log-corrections has been discussed. A quantitative study of the log corrections goes beyond the scope of this manuscript and would require more complicated model with longer range interactions along the teeth and the backbone as pointed by the referee. Indeed, longer-range couplings, for example a J1-J2 model along the teeth at J2/J1=0.2411, can tune the marginal operator to zero and suppress the log-corrections for weak backbones and L-even. It is less clear though how to remove the corrections associated with the weakening of the boundary conditions that appears upon increasing the backbone interactions (see for example the curve for Jbb=0.5 in Fig.8(a)) without affecting the backbone too much. More puzzling in this respect is the absence of log-corrections for L-odd and Jbb<<1 and almost excellent agreement with linear scaling with 1/(NL) (see Fig.6). Either the corrections along the tooth and along the backbone have opposite signs and compensate each other or there is an effective interaction between the delocalized spins; we do not know yet. The exploration continues.

All the remaining comments from referees were helpful and the manuscript has been modified accordingly.

---

## Round 2 · List of Changes

1. The definition of the entanglement entropy has been included along with a sketch of the bi-partition of a comb.
  2. The discussion on CFT predictions for Friedel oscillations has been extended and relevant references have been included.
  3. Eq.4 has been re-written and all parameters entering the equation have been defined.
  4. We have added an extended discussion about the CFT prediction for the structure of the excitation spectrum.
  5. We have significantly extended the discussion of log-corrections.

---

## Editorial Decision

published